# Preparation and Application of Organic-Inorganic Nanocomposite Materials in Stretched Organic Thin Film Transistors

**DOI:** 10.3390/polym12051058

**Published:** 2020-05-05

**Authors:** Yang-Yen Yu, Cheng-Huai Yang

**Affiliations:** 1Department of Materials Engineering, Ming Chi University of Technology, New Taipei City 243, Taiwan; harry830308@gmail.com; 2Department of Chemical and Materials Engineering, Chang Gung University, Taoyuan City 33302, Taiwan

**Keywords:** soluble polyimide, polyurethane, Jeffamine, organic-inorganic hybrid film, Stretchable transistor

## Abstract

High-transparency soluble polyimide with COOH and fluorine functional groups and TiO_2_-SiO_2_ composite inorganic nanoparticles with high dielectric constants were synthesized in this study. The polyimide and inorganic composite nanoparticles were further applied in the preparation of organic-inorganic hybrid high dielectric materials as the gate dielectric for a stretchable transistor. The optimal ratio of organic and inorganic components in the hybrid films was investigated. In addition, Jeffamine D2000 and polyurethane were added to the gate dielectric to improve the tensile properties of the organic thin film transistor (OTFT) device. PffBT4T-2OD was used as the semiconductor layer material and indium gallium liquid alloy as the upper electrode. Electrical property analysis demonstrated that the mobility could reach 0.242 cm^2^·V^−1^·s^−1^ at an inorganic content of 30 wt.%, and the switching current ratio was 9.04 × 10^3^. After Jeffamine D2000 and polyurethane additives were added, the mobility and switching current could be increased to 0.817 cm^2^·V^−1^·s^−1^ and 4.27 × 10^5^ for Jeffamine D2000 and 0.562 cm^2^·V^−1^·s^−1^ and 2.04 × 10^5^ for polyurethane, respectively. Additives also improved the respective mechanical properties. The stretching test indicated that the addition of polyurethane allowed the OTFT device to be stretched to 50%, and the electrical properties could be maintained after stretching 150 cycles.

## 1. Introduction

Stretchable electronic components have attracted much research interest due to their considerable potential in biomedical instruments, smart skins, displays, and battery devices [1,2,3]. From 2010 to 2020, thin film transistors have been made predominantly from inorganic materials. The main reason for this is that the carrier mobility values of organic materials are too low compared with those of inorganic materials [4]. Because the performance of an organic thin film transistor (OTFT) [5,6] has not been able to reach the same performance of inorganic transistor, researchers have continued to study the use of various semiconductor materials [7,8,9,10,11] to improve their carrier mobility. In addition, the plastic soft board-based OTFTs can also be used on flexible substrates [12,13,14,15]. The rise of plastic substrates [16,17,18] has necessitated some flexural quality measurements and novel processing methods, such as stretching and coating, to increase the flexibility and mobility of components [19,20]. A roll-to-roll process that can be fabricated on a flexible substrate in a low-temperature environment could support future commercial development [21].

Hybrid materials [22,23,24] are organic-inorganic polymer blends that are molecularly mixed and blended through van der Waals forces, hydrogen bonds, ionic bonds, or covalent bonds, thus overcoming the phase separation that can usually be observed in traditional materials. These hybrid materials have the advantages of organic and inorganic materials, providing excellent material properties, including thermal, mechanical, optical, and electrical properties. To achieve a good nanoscale dispersion of organic-inorganic materials, the sol-gel method is the most commonly used method because it is flexible and materials prepared with the sol-gel method have high thermal stability and optical transparency.

This study used spin coating to replace the traditional vaporization for the fabrication of thin film transistors. Polyimide [25,26,27] was used for the preparation of OTFT due to its good thermal stability, chemical resistance, and mechanical properties. As practical applications continue to advance, the requirements for thermal and mechanical properties are becoming more and more demanding, so inorganic materials are often used to enhance the relevant properties. The most common inorganic materials are SiO_2_ and TiO_2_, which can be prepared using tetraethoxysilane (TEOS) and titanium butoxide, respectively. The use of inorganic composite material TiO_2_-SiO_2_ has also been featured in the literature [28,29]. Such an organic-inorganic hybrid film [30] was applied in OTFT as a dielectric film. The donor material, PffBT4T-2OD [31], was also used to replace the traditional pentacene [32] as a semiconductor layer in organic photovoltaic devices. However, PffBT4T-2OD has not been applied to OTFTs in other research.

Electronic products increasingly require the properties of flexibility and stretchability [33,34,35,36,37]. Therefore, some suitable polymers have been added to these advanced electronic products [38,39], such as Jeffamine D2000 [40] and polyurethane [41]. Another approach to enhance flexibility and stretchability is to connect the sidechain of the semiconductor layer material with an elastic polymer, such as poly(butyl acrylate)(PBA) or 2,6-pyridine dicarboxamide (PDCA). After this modification, the researchers expected that the device could retain its original performance after being subjected to stretching many cycles. The chemical structures of the polyimide-TiO_2_-SiO_2_, Jeffamine D2000, and polyurethane as well as the structural diagrams for the OTFT device and the experimental stretching directions are shown in Figure 1. Tensile properties depend on the ratio of TiO_2_-SiO_2_ and whether Jeffamine D2000 or polyurethane is added. The addition ratio of TiO_2_-SiO_2_ ranges from A0–A40 in the order of 0 wt.% to 40 wt.%, B0-B40 when Jeffamine D2000 is added, and C0–C40 when polyurethane is added.

## 2. Experimental Section

In this study, a stretchable OTFT was fabricated using Elastomer Tape 3M tape as the stretchable substrate and poly(3,4-ethylenedioxythiophene) polystyrene sulfonate (PEDOT:PSS, Sigma Aldrich, Darmstadt, Germany) as the lower electrode. TiO_2_-SiO_2_ inorganic nanoparticles and a soluble polyimide with COOH and a fluorine atom functional group were used to prepare the dielectric layer. The COOH on polyimide could be hydrolyzed and condensed with TiO_2_-SiO_2_ to form a dense network structure, and the size of the CF group in PI molecule is quite big, which can cause an increase of free volume and a reduction of the interaction between the molecular chains, so as to increase the solubility and transparency for the prepared polyimide-TiO_2_-SiO_2_ hybrid films. The film was used as an OTFT gate dielectric. In addition, soluble polyimide overcame the problem of the high temperature dehydration cyclization of thermal polymerization and was applicable to a stretchable OTFT device. In addition, Jeffamine D2000 and polyurethane could be used as additives to increase the tensile properties without the original electrical properties being affected.

### 2.1. Preparation of Dielectric Gate Dielectric

Briefly, the 4,4-oxydiphthalic anhydride (97%, Sigma Aldrich, Darmstadt, Germany) and the 2,2-*bis*(3-amino-4-hydroxyphenyl) hexafluoropropane (98%, Matrix Scientific, Columbia, SC, USA) in a three-necked flask were dissolved in the *n*-methyl-2-pyrrolidone (NMP, 99.9%, TEDIA, USA) with 1:1 molar ratio and mixed uniformly. After the further addition of isoquinoline (95%, Tokyo Chemical Industry) in a nitrogen atmosphere for 5 h, a yellow-brown solution was obtained, which was poly (amic acid) (PAA). The PAA was placed in an oil bath at 150 °C for 18 h. The polyimide solution obtained was placed in a water: methanol (98%, Mallinckrodt Baker, Phillipsburg, KS, USA) (1:3) mixed solvent to produce the precipitate. The filtrated precipitate was placed in a vacuum oven and dried at 60 °C for 2 days to obtain a soluble polyimide powder containing COOH and a fluorine functional group. Tetraethyl orthosilicate (TEOS, Sigma Aldrich, Darmstadt, Germany) was dissolved in ethanol (99.5%, Acros Organics, NJ, USA) added to an aqueous solution of nitric acid, and stirred for 30 min. Simultaneously, titanium(IV) butoxide (Ti(OBu)_4_, Sigma Aldrich, Darmstadt, Germany) was dissolved in 2-methyl-2,4-pentanediol (98%, Alfa Aesar, MA, USA) solvent, stirred for 30 min. The two aforementioned solutions were mixed and stirred for sol-gel reaction for 30 min, and the solvent was then removed with a rotary evaporator and finally placed in an oven to obtain the TiO_2_-SiO_2_ inorganic nanoparticles. The polyimide dissolved in N,N-Dimethylacetamide (DMAc, 99.8%, TEDIA, USA) and the TiO_2_-SiO_2_ nanoparticles dispersed in butanol solvent were mixed and stirred for 30 min to prepare three series of hybrid materials, namely polyimide-TiO_2_-SiO_2_, polyimide-TiO_2_-SiO_2_:D2000, and polyimide-TiO_-_SiO_2_:PU.To prepare the polyimide-TiO_2_-SiO_2_ hybrid material, the different ratios of SiO_2_-TiO_2_ (0, 10, 20, 30, and 40 wt.%) were mixed with polyimide and stirred for 1 h to obtain the PI-TiO_2_-SiO_2_ precursor solution represented by AX (X = weight percentage of SiO_2_-TiO_2_ in hybrid material). For preparation of the polyimide-TiO_2_-SiO_2_:D2000 and polyimide-TiO_2_-SiO_2_: PU hybrid material, the preparation procedure was the same as for polyimide-SiO_2_-TiO_2_. The only difference was that the polymer (Jeffamine D2000, Alfa Aesar, Massachusetts, USA) or polyurethane, (Sigma Aldrich, Darmstadt, Germany) was dropped gradually into the polyimide solution before the mixing with TiO_2_-SiO_2_ inorganic nanoparticles. The polyimide-TiO_2_-SiO_2_:D2000 and polyimide-TiO_2_-SiO_2_: PU hybrid materials were represented by BX and CX, respectively, where X was the weight proportion of SiO_2_-TiO_2_ in the hybrid material.

### 2.2. OTFT Device Preparation

First, the elastomer tape was attached to the glass and subjected to plasma treatment for 3 min to clean the tape surface. PEDOT:PSS was then spin coated on the elastomer tape and annealed at 100 °C for 30 min. The solution of polyimide-SiO_2_-TiO_2_ (or polyimide-SiO_2_-TiO_2_:Jeffamine D2000 or polyimide-TiO_2_-SiO_2_:polyurethane) was spin coated onto PEDOT:PSS elastomer tape at 2000 rpm/20 s. The coated wafer was placed on a hot plate and thermally polymerized through stepwise heating. The baking process was performed at 60, 80, and 100 °C for 10 min and then, finally, at a temperature of 120 °C for 10 min. Three series of hybrid dielectric films, namely AX, BX, and CX, were obtained. The poly[(5,6-difluoro-2,1,3-benzothiadiazol-4,7-diyl)-alt-(3,3’’’-di(2-octyldodecyl)-2,2’,5’,2’’,5’’,2’’’-quaterthiophen-5,5’’’-diyl) (PffBT4T-2OD, Sigma Aldrich) as the active layer was then spin coated onto the dielectric layer on a hot plate and heated at 90 °C for 5 min as an annealing process. The upper electrode (source and drain) EGaIn (99.99%, Alfa Aesar, MA, USA) was dropped onto the lower electrode and the PffBT4T-2OD surface, respectively, to fabricate the OTFT device. The device structure is shown in Figure 1.

### 2.3. Characterization 

The thermal properties of the prepared hybrids were assessed using a thermogravimetric analysis (TGA, TA Instruments, Q50) and differential scanning calorimeter analysis (DSC, TA Instruments, Q20/RSC90) at heating rates of 20 °C and 10 °C/min, respectively. The transmittances of the hybrid films coated on the quartz substrates were collected using an ultraviolet-visible spectrum (UV-Vis, Jasco, V-650). The morphologies of the thin films were observed with a high-resolution transmission electron microscope (HR-TEM, JEOL, JEM-2100), a scanning electron microscope (SEM, Hitachi, H-2400), and an atomic force microscope (AFM, Veeco, DI 3100). The thicknesses of the hybrid thin films were analyzed with a microfigure measuring instrument (Surface Profiler, *α*-*step*, ET-4000, Kosaka Laboratory Ltd.). For the metal-insulator-metal (MIM) structure analysis, 0.6-mm diameter Al electrodes were deposited directly onto the gate dielectric films through shadow masking. MIM direct current measurements and OTFT measurements were performed in ambient conditions using a probe station interface with an Agilent E4980A precision LCR meter (10 kHz to 1 MHz) and an Agilent B1500A semiconductor device parameter analyzer. 

## 3. Results and Discussion

Figure 1 shows the chemical structures of the polyimide-TiO_2_-SiO_2_ composite dielectric material, Jeffamine D2000, and polyurethane additives and the schematic for the OTFT device structure and the tensile direction. The OTFT devices exhibit tensile properties that depend on the addition ratio of TiO_2_-SiO_2_ inorganic nanoparticles and the presence or absence of Jeffamine D2000 or polyurethane additives. The addition ratio of TiO_2_-SiO_2_ inorganic nanoparticles ranges from 0 wt.% to 40 wt.%; the cases with those ratio values are denoted by A0–A40, B0–B40, and C0–C40, respectively, indicate the addition of Jeffamine D2000 and polyurethane additives in the order of 0 wt.% to 40 wt.%.

### 3.1. Analysis of Optical and Thermal Properties

All of the prepared hybrid films have optical transmittances greater than 90% with the film thickness about 200 nm. Figure 2a shows the UV-vis spectra of the optical transmittance of A0, A30, B30, and C30 hybrid thin films in the visible light region of 400–700 nm, the optical transmittance is greater than 90%. This result shows that the composite dielectric film has good transparency (as listed in Table 1). Appendix A shows the TEM image of inorganic TiO_2_-SiO_2_ nanoparticles. It shows that the size of the particles of the as-prepared TiO_2_-SiO_2_ is about 30–40 nm with a spherical morphology. When the particle size is less than 50 nm, light scattering can be negligible [42]. Moreover, Appendix A shows the optical transmittance of A0, A30, B30, and C30 films as the dielectric layer of OTFTs device in the visible light region of 400–700 nm. This indicates that the optical transmittances of all samples are greater than 75%. The thermal properties of the prepared polyimide-TiO_2_-SiO_2_ composite dielectric films were analyzed by thermogravimetric analysis (TGA). Figure 2b shows the TGA curves undertaken in a nitrogen atmosphere. It reveals that the decomposition temperature (T_d_) of A0, A30, B30, and C30 hybrid thin films are 418, 450, 461, and 443 °C, respectively. The relative parameters for thermal properties are listed in Table 1. This indicates that the thermal decomposition temperature increases with the content of TiO_2_-SiO_2_ nanoparticles due to the formation of chemical bonding between polyimide and TiO_2_-SiO_2_, which can restrict the polyimide chain reaction, and the T_d_ and thermal stability for the hybrid films thus increases as TiO_2_-SiO_2_ content increases [22]. In addition, the addition of Jeffamine D2000 and polyurethane also increase the T_d_ from 426 °C to 477 °C for B0–B40 and 405 °C to 454 °C for C0–C40. The increase in T_d_ for B0–B40 is due to the hydrogen bonding between the N atom in the Jeffamine D2000 and the composite dielectric material. However, the polyurethane is a softer polymer, so the T_d_ of B0–B40 is expected to be lower than that for the other two series of hybrid films. However, the T_d_ for all of hybrid films nonetheless exceed 400 °C, indicating good thermal stability. In addition, none of the hybrid films exhibit weight loss at temperatures lower than 300 °C, and the residual quantity of A0–A40 increased with increasing quantities of TiO_2_-SiO_2_ added when the temperature increased to 900 °C. At 900 °C most of the polyimide has completely decomposed, and the remaining residual quantity is an inorganic oxide forming a cross-linked stable network. This result demonstrates that inorganic TiO_2_-SiO_2_ nanoparticles have been successfully incorporated into organic materials. Figure 2c shows the differential scanning calorimeter analysis (DSC) measured in a nitrogen atmosphere. It reveals that the glass transition temperatures (T_g_) of A0 (PI), B0 (PI:D2000), and C0 (PI:PU) are 266 °C, 286 °C, and 270 °C, respectively. In addition, no T_g_ point of all samples can be observed in Figure 2c in the temperature range of 25–350 °C, showing the T_g_ of all hybrid materials (PI/TiO_2_-SiO_2_) prepared in this study exceeds 350 °C (as listed in Table 1). It is known that the inorganic TiO_2_-SiO_2_ nanoparticles can be uniformly distributed in the polyimide matrix, and form a crosslinking structure between the polyimide and nanoparticles, which restricts the chain motion and strengthens the polyimide strength, thereby causing an increase in T_g_ and T_d_. The results of thermal analysis suggest that all of the hybrid films prepared in this study exhibit good heat resistance and no phase separation between the polyimide and TiO_2_-SiO_2_ nanoparticles [43].

### 3.2. Analysis of Stretching Properties

Figure 3 shows the optical microscopy images of (a) A0, B0, and C0 and of (b) A30, B30, and C30 thin films subject to various strain levels. Figure 3a shows that bare polyimide film generates cracks when subjected to a stretching ratio of 30%, and numerous cracks and wrinkles are produced when the stretching ratio reaches 50%. The B0 film exhibits only wrinkles under the stretching ratio of 30%, but numerous cracks are also present as the stretching ratio increases to 50%. Compared with the result for A0, the addition of Jeffamine D2000 (B0) is seen to improve the film’s stretchability. For the C0, the film shows no cracks or wrinkles subject to stretching ratios of 30% and 50%. The results reveal that the addition of polyurethane is more effective than the addition of Jeffamine D2000 for improving the tensile properties of the thin films in the absence of inorganic particles in the polyimide matrix. Moreover, the effect of nanoparticles TiO_2_-SiO_2_ on the tensile properties is seen in Figure 3b. Adding TiO_2_-SiO_2_ is seen to cause the tensile properties of the A30 films to decrease obviously because more cracks are observed for a stretching ratio of 30%. For the case of B30 films, wrinkles continue to be generated at a stretching ratio of 30%, but no cracks are observed. The C30 films’ stretchability is optimal at the stretching ratios of 30% and 50%. No cracks or wrinkles are observed on the C30 films. Therefore, from the optical microscopy diagram of these dielectric hybrid films, the addition of Jeffamine D2000 and polyurethane polymers is seen to increase the film stretchability and the addition of polyurethane produces tensile properties superior to those obtained from the addition of Jeffamine D2000. 

### 3.3. Analysis of Surface Morphology and Surface Energy

The surface flatness of the hybrid dielectric film s was measured using an atomic force microscope (AFM). The AFM result demonstrates that the surface roughness (Ra) of the three series of A0–A40, B0–B40, and C0–C40 are 0.44–0.75, 0.31–0.64, and 0.43–0.69 nm, respectively. Ra increases with the increase in TiO_2_-SiO_2_ content. However, all hybrid dielectric films were produced without pinholes, and the surface flatness (the ratio of Ra to film thickness, Ra/h) for all hybrid films was less than 0.35% (Table 1), indicating that all prepared hybrid dielectric films in this study had a good surface flatness. In the previous studies, it has been confirmed that, when the ratio of surface roughness to thickness is less than 0.5%, the material has a good flatness. The prepared dielectric thin film in this study has a better surface flatness than those of studies in literature [39,42,43]. According to the aforementioned results, the three series of hybrid dielectric films have good light transmission, thermal stability, and surface flatness, and no phase separation was observed. Therefore, the prepared hybrid dielectric films can be effectively applied as the gate dielectric materials for the OTFT application.

Typically, thin films show a light scattering behavior due to the surfaces roughness. An innovative hybrid thin film with a lower-than-usual surface roughness can reduce the light loss on the waveguide surface. This result also confirms the potential for the use of polyimide-TiO_2_-SiO_2_ hybrid material in OTFT as the dielectric film. The surface flatness of the dielectric layer greatly influences the characteristics of the OTFT. When the dielectric layer has a low surface roughness, it effectively reduces the leakage current of OTFT and promotes the order growth of crystals in the active layer. Figure 4 shows the AFM images of the semiconductor layer (BffBT4T-2OD) coated on the various dielectric composite films, namely A0, B0, C0, A30, B30, and C30. This indicates that island-like aggregation was produced from the BffBT4T-2OD on the semiconductor layer (A30) after the addition of the inorganic nanoparticles, and the degree of aggregation became more obvious after the addition of the Jeffamine D2000 (B30) and polyurethane (C30). These dense aggregates of BffBT4T-2OD can help to increase the tensile properties of these composite films and the stretchability of OTFT [41]. BffBT4T-2OD is hydrophobic in nature, and we observed B30 to have lower surface energy (42.07 mJ·m^−2^) than A30 and C30 did, which enables the growth of BffBT4T-2OD. As the adding ratio of the TiO_2_-SiO_2_ nanoparticles increases, the size of BffBT4T-2OD crystal grains in the semiconductor layer also increases, resulting in better characteristics for the OTFT. This may be related to the affinity of the dielectric surface for BffBT4T-2OD. As the TiO_2_-SiO_2_ content increases from 0% (A0) to 30 wt.% (A30), the surface energy of the dielectric layer decreases, which causes the grain size of BffBT4T-2OD in semiconductor layer to increase. The well-connected domain of the A30 provides an efficient channel for charge transport and increases the charge carrier density at the interface between dielectric and semiconductor, which can prevent the charge defects from occurring at the interface and improve the performance of the OTFTs [38,39,40]. 

The contact angles of the gate dielectric films were measured using deionized water and diiodomethane as the test drops, respectively, due to their different polarities. The surface energies of the polyimide-TiO_2_-SiO_2_ hybrid films could then be calculated from the values of the contact angles obtained from water and diiodomethane drops, respectively. The results are listed in Table 2. If the surface of a hybrid film has a small water contact angle, this indicates that the surface is hydrophilic and has a large surface energy. Conversely, a large water contact angle indicates that the surface is hydrophobic and has a low surface energy. As shown in Table 2, whether water or diiodomethane is used as the test drop, the contact angles for the hybrid films (A, B, and C) increase first and then decrease with the addition of inorganic particles. For A0–A40 hybrid films, the water contact angles increase from 79.78 for A0 to 83.99 for A30 and then decrease to 78.30 for A40. 

The increase in water contact angle is mainly due to the change in surface roughness of hybrid films. In addition, the PI/SiO_2_-TiO_2_ hybrid films in this study have a polarizable and weakly hydrophobic surface, resulting in this dielectric layer having a low surface energy. This is a very important property for wetting of the latter deposited organic semiconductor layer, and can improve the performance of the device [44]. Therefore, when 30 wt.% of TiO_2_-SiO_2_ was added, the surface energy of the hybrid film was lowered from 51.02 mJ·m^−2^ (A0) to 44.17 mJ·m^−2^ (A30), indicating that the addition of high dielectric TiO_2_-SiO_2_ in a low dielectric PI matrix can change the surface roughness of the PI/TiO_2_-SiO_2_ hybrid film, which in turn affects the surface energy of the hybrid film. In addition, due to the inherent hydrophobic nature of the polymer, the addition of Jeffamine D2000 and polyurethane additives can produce a highly hydrophobic surface, which further reduces the surface energy. The results show that the lowest surface energy obtained from B30 is 42.07 mJ·m^−2^ [31]. Typically, dielectric surfaces with low surface energy can provide a venue for the growth of organic semiconductor chains. 

### 3.4. Analysis of Electrical Properties

The result of volumetric capacity measurement shows that the capacitance (at 1 kHz–1 MHz) increases as the TiO_2_-SiO_2_ ratio increases, and the relationship is linear. At lower frequencies, the capacitance may increase slightly due to the increased response time available for polarization. The dielectric constant (k) is evaluated using the following equation:(1)$$C=\frac{{k\varepsilon }_{0}A}{d}$$
where C is the measured capacitance, ε_0_ is the vacuum dielectric constant, A is the area of the capacitor, and d is the thickness of the dielectric layer. As shown in Table 2, the dielectric constants obtained at 1 kHz were 4.53 for A0, 9.51 for A40, 4.47 for B0, 8.58 for B40, 4.22 for A0, and 7.87 for A40. When a higher concentration of TiO_2_-SiO_2_ was used in a film’s fabrication, its dielectric constant was higher. Moreover, the electrical data of OTFTs fabricated by various hybrid dielectrics are shown in Table 3. The results show that the values of mobility (μ) and switch current ratio (on-off current ratios, I_on_-I_off_) increase with increasing the TiO_2_-SiO_2_ content. Moreover, the leakage current density (LCD) measured at −2 MV·cm^−1^ also increases as the content of TiO_2_-SiO_2_ nanoparticles increases. Appendix A. shows the transfer curves of OTFTs prepared by different dielectric materials, A0, A30, B30, and C30. This indicates that the mobility and I_on_/I_off_ of the device prepared by the polyimide dielectric layer without TiO_2_-SiO_2_ nanoparticles (A0) are 0.0181 cm^2^·V^−1^·s^−1^ and 1.13 × 10^3^, respectively. When the PI/TiO_2_-SiO_2_ hybrid material (A30) was used as a dielectric layer, the mobility and I_on_/I_off_ of device increased to 0.242 cm^2^·V^−1^·s^−1^ and 9.04 × 10^3^. When adding Jeffamine D2000 (B30) and polyurethane (C30) additives into the dielectric layer, the mobility and I_on_/I_off_ were further increased to 0.817 cm^2^·V^−1^·s^−1^ and 4.27 × 10^5^ for B30 and 0.562 cm^2^·V^−1^·s^−1^ and 2.04 × 10^5^ for C30, respectively, which shows that the proper amount of TiO_2_-SiO_2_ nanoparticles and additives can effectively improve the device performance. The larger the I_on_/I_off_ ratio, the better the switch characteristics of OTFTs. It is known that high-k dielectric layer could cause a low operation voltage and low power consumption. The LCD value is related to the thickness of the hybrid dielectric layer and the pinholes density, because the chemical bonding between the inorganic and polyimide can make the dielectric layer structure more dense, and the high-k dielectric can improve the capacitive coupling effect between the gate and active channel layer, which can increase the driving current and reduce the operating voltage. In addition, the surface morphology of the dielectric layer affects the structure of the deposited organic semiconductor layer, which in turn affects the performance of the device. A smooth and pinhole-free surface of dielectric layer is important for the interfacial connection during the deposition of organic semiconductor layers, because a smooth interface reduces the charge scattering sites in the channel. Reducing the current leakage from the dielectric interface has a great influence on the electrical performance of the OTFTs. For high-performance OTFTs, the gate dielectric should have a low LCD and a high-k, which can provide greater surface charge accumulation and simultaneously reduce the operating voltage. It should be noted that the thickness of the dielectric layer must be carefully controlled to minimize the current leakage without greatly reducing the capacitance [44]. An increase in the LCD value means that the effect of the insulating layer is reduced. As shown in Table 3, the LCD values of all dielectric layers were less than 10^−8^ A·cm^−2^. Therefore, these hybrid dielectric layers are suitable for OTFT applications. The gate-current behavior is usually similar to the capacitor leakage current. In this work, we used a metal-insulator-metal capacitor to study the dielectric leakage current. Moreover, as the surface energy decreases, the particle size and alignment of the semiconductor layer become denser, and the carrier mobility of the OTFT increases. Namely, more hydrophobic material increases the carrier mobility of the OTFT. Therefore, the increase in inorganic content helps to form a good organic polymer film, reducing the structural defects in the film and increasing compactness, thus improving carrier mobility. However, the device mobility decreases when the TiO_2_-SiO_2_ content is more than 30 wt.%, which may be attributed to the coarser surface and the aggregation of the TiO_2_-SiO_2_ particles, disturbing the formation of BffBT4T-2OD crystal structure in the semiconductor layer. Appendix A shows the output characteristics of the OTFTs using (a) (A0), (b) A30, (c) B30 and (d) C30 as the dielectric layer, respectively. The threshold voltages (V_t_) of OTFTs based on hybrid films are small, so only a small gate voltage is needed to turn on the gate. Surface polarization may result in smaller threshold voltages, which can lead to the filling defects of local carrier. Most of the V_t_ displacement is affected by three factors, which are the charge defect trapping, surface polarization, and ions. In this study, the variation of V_t_ displacement might be attributed to the addition of different proportions of inorganic nanoparticles at the interface between the dielectric and the semiconductor layers [45,46].

Figure 5a shows the mobility values of A0, B0,C0, A30, B30, and C30 at various strain values. These results prove that devices with Jeffamine D2000 (B0, B30) and polyurethane (C0, C30) as additives can be stretched to 20% and 50%, respectively. Figure 5b shows the mobility of A0, B0, C0, A30, B30, and C30 at various stretch cycles, indicating that the devices with Jeffamine D2000 (B0, B30) and polyurethane (C0, C30) as additives can be stretched up to 150. The aforementioned results show that the devices with polyurethane additive can achieve superior stretching properties of 50% for stretching 150 cycles. The mobility has almost no change, which is because the polyurethane additive is a softer chain polymer. AFM analysis revealed that the hybrid films with polyurethane always exhibit a denser and more concentrated film structure that is advantageous for the stretching propertes of the stretchable devices. In addition, the devices with Jeffamine D2000 can also achieve a good stretching properties of 50% for stretching 150 cycles. Although the mobility is reduced by approximately 10%. However, the A0 and A30 samples without added any additives have a significant problem in that the mobility decreases sharply and does not have stretchability.

Finally, we applied three series of dielectric materials (A, B, C) in the OTFT device as gate materials. Table 3 summarizes the electrical characteristics of these OTFTs, including the LCD, mobility, and I_on_-I_off_. As mentioned, the electrical properties of the fabricated OTFT devices are identical to those obtained using AFM and surface energy. In the upstretched case, the LCD becomes larger with the increase in the content of TiO_2_-SiO_2_ inorganic nanoparticles, but the LCD value is less than 10^−8^ A·cm^−2^ (−2 MV·cm^−1^) (Figure 5c). The addition of Jeffamine D2000 and polyurethane additives reduces the LCD values. The mobility and I_on_-I_off_ of the dielectric layers of pure polyimide without inorganic nanoparticles and polymer additives (A0, B0, and C0) are 1.81 × 10^−2^ cm^2^·V^−1^·s^−1^ and 1.13 × 10^3^ for A0, 5.04 × 10^−2^ cm^2^·V^−1^·s^−1^ and 1.51 × 10^4^ for B0, and 4.81 × 10^−2^ cm^2^·V^−1^·s^−1^ and 8.13 × 10^3^ for C0, respectively. The mobility and I_on_-I_off_ increase obviously with the content of TiO_2_-SiO_2_ inorganic nanoparticles. The mobility and I_on_-I_off_ of A30 hybrid film reach 2.42 × 10^−1^ cm^2^·V^−1^·s^−1^ and 9.04 × 10^3^, respectively. Moreover, after the addition of Jeffamine D2000 (B30) and polyurethane (C30), the mobility and I_on_-I_off_ improve to 8.17 × 10^−1^ cm^2^·V^−1^·s^−1^ and 4.27 × 10^5^, respectively, for B30 and to 5.62 × 10^−1^ cm^2^·V^−1^·s^−1^ and 2.04 × 10^5^, respectively, for C30. However, when the content of SiO_2_-TiO_2_ increases to 40 wt.% (A40, B40, and C40), the mobility and switching current ratio decrease to 1.07 × 10^−2^ cm^2^·V^−1^·s^−1^ and 1.16 × 10^3^, respectively, for A40 and to 5.51 × 10^−1^ cm^2^·V^−1^·s^−1^, and 8.50 × 10^4^, respectively, for B40, and to 2.07 × 10^−1^ cm^2^·V^−1^·s^−1^ and 2.16 × 10^3^, respectively, for C40. 

The decreases in mobility and switching current ratio are attributed to the phase separation of the mixed solution when the content of inorganic particles of TiO_2_-SiO_2_ is too high. The results show that the use of TiO_2_-SiO_2_ inorganic nanoparticles and Jeffamine D2000 and polyurethane additives can improve the mobility and I_on_-I_off_. However, when the content of TiO_2_-SiO_2_ inorganic nanoparticles reaches 40 wt.%, precipitation and subsequent phase separation occurs in the precursor solution, resulting in poor film properties and thus poor electrical properties for A40, B40, and C40 samples. Therefore, the optimal mass ratio of polyimide and TiO_2_-SiO_2_ inorganic nanoparticles is 70:30 wt.%. The aforementioned results demonstrate that the content of TiO_2_-SiO_2_ nanoparticles exert an obvious influence on the electrical performance of OTFTs. In summary, the mobility and current-switching ratio of B30- and C30-based OTFT after being stretched 150 cycles at 50% of strain are 0.29 cm^2^·V^−1^·s^−1^ and 8.17 × 10^4^ for B30-based OTFT and 0.38 cm^2^·V^−1^·s^−1^ and 1.34 × 10^5^ for C30-based OTFT, respectively, which is higher than those obtained from other hybrid dielectrics-based devices. These results show that Jeffamine D2000 additives (B30) and polyurethane additives (C30) improve the properties of stretchable OTFE devices, of which C30 exerts the better effect. This result is consistent with the optical microscopy result of the tensile test for the hybrid dielectric films.

## 4. Conclusion

In this study, we successfully synthesized a series of hybrid dielectric films using polyimide and SiO_2_-TiO_2_ nanoparticles without polymer additives, namely polyimide-TiO_2_-SiO_2_, polyimide-TiO_2_-SiO_2_:D2000, and polyimide-TiO_2_-SiO_2_:PU. Pffbt4t-2OD was used in the semiconductor layer. The addition of Jeffamine D2000 (D2000) and polyurethane (PU) as additives was observed to increase the tensile properties without affecting the original electrical properties. The results suggest that the C30-based OTFT achieves the best tensile effect of 50% train after 150 cycles subject to the 10% mobility reduction because the polyurethane polymers are softer and can provide a denser and more concentrated film structure, which facilitates the stretching of the device. Through the adjustment of the ratios of various TiO_2_-SiO_2_ inorganic nanoparticles, the dielectric constant of the hybrid material can be adjusted, thereby significantly improving the dielectric properties of the dielectric layer. The device properties (mobility and threshold voltage) and film properties (dielectric constant, surface morphology, and hydrophilic hydrophobicity) exhibit a strong correlation to the proportion of TiO_2_-SiO_2_ inorganic nanoparticles. This study shows that the prepared hybrid films can be customized according to the requirements for practical applications. In addition, our PI-hybrid material has the advantage of transparency, high thermal stability, and environmental safety. The addition of Jeffamine D2000 and polyurethane can increase tensile properties without affecting the original electrical properties and widen the applicability of OTFT devices. 

## Figures and Tables

**Figure 1 polymers-12-01058-f001:**
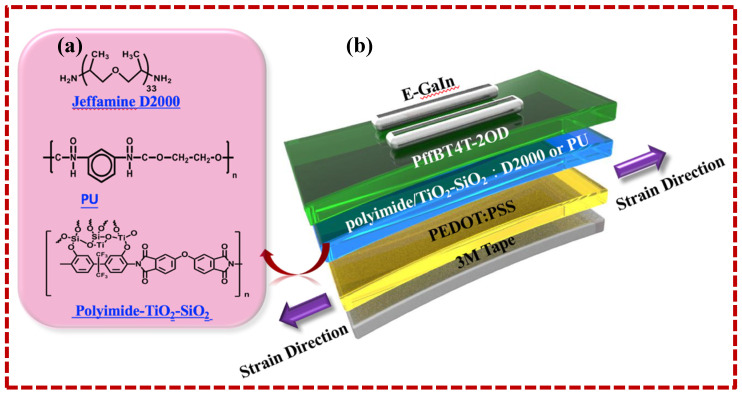
(**a**) Chemical structures of polyimide-TiO_2_-SiO_2_, Jeffamine D2000, and PU. (**b**) Device structure with illustration of each layer and strain direction in an organic thin film transistor.

**Figure 2 polymers-12-01058-f002:**
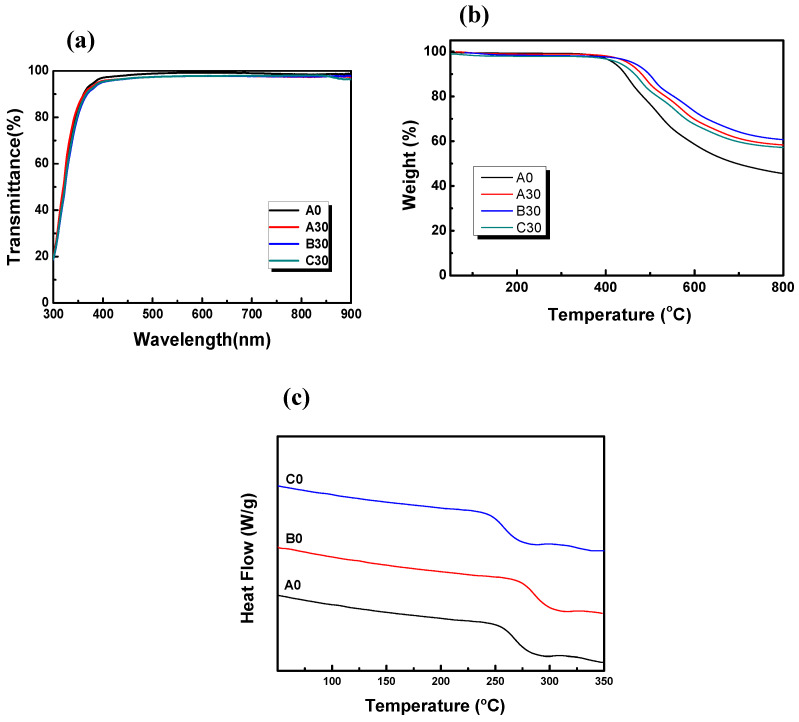
(**a**) UV-vis spectra of the optical transmittance, (**b**)TGA curves, and (**c**) DSC curves of hybrid thin films.

**Figure 3 polymers-12-01058-f003:**
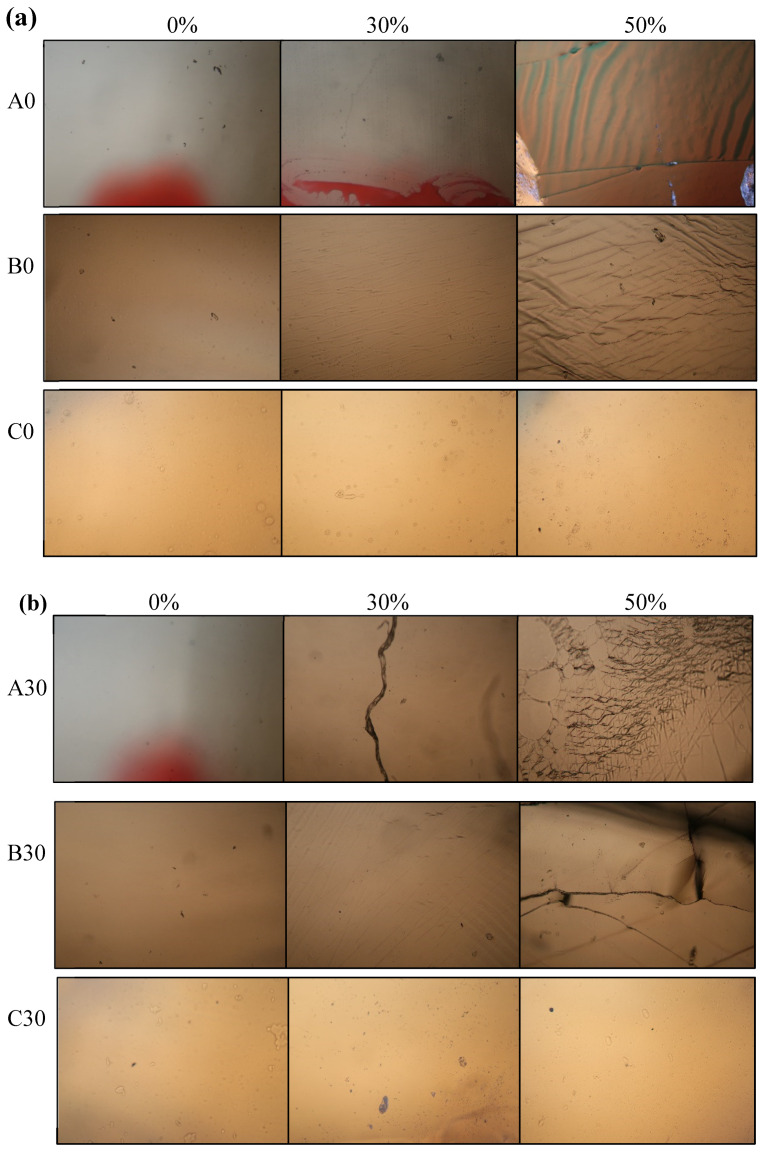
Optical microscopy images of (**a**) A0, B0, and C0 and of (**b**) A30, B30, and C30 thin films subject to various strain levels.

**Figure 4 polymers-12-01058-f004:**
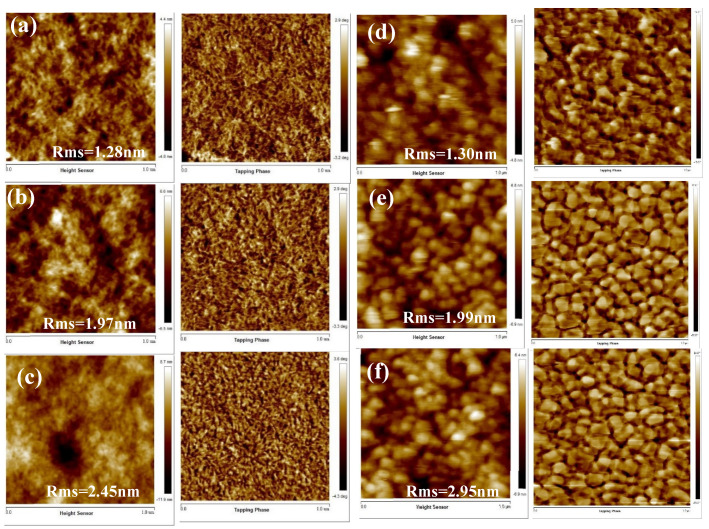
Tapping-mode AFM (1 × 1 μm) images (left: topographic images, right: phase images) of blend films deposited with various TiO_2_-SiO_2_ ratios and addition of Jeffamine D2000 or polyurethane; (**a**) A0, (**b**) B0, (**c**) C0, (**d**) A30, (**e**) B30, and (**f**) C30.

**Figure 5 polymers-12-01058-f005:**
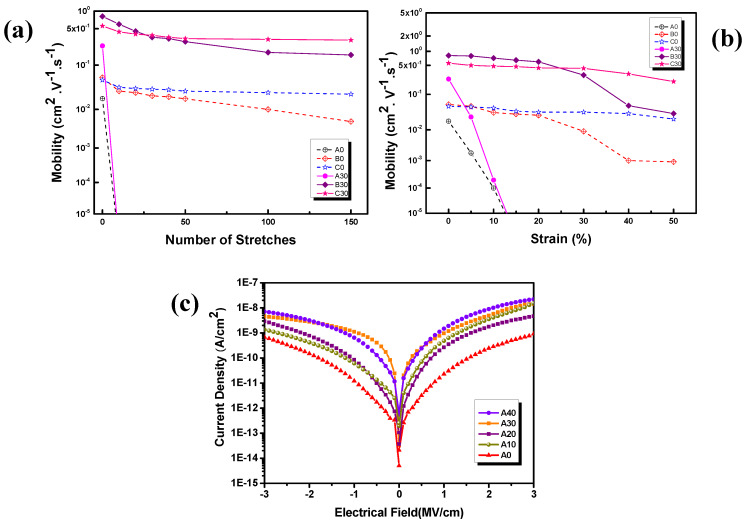
(**a**) mobility of the strained percentage, (**b**) the strained cycles and (**c**) leakage characteristics of hybrid thin films.

**Table 1 polymers-12-01058-t001:** Summary of properties of the prepared dielectric hybrid film.

No.	H (nm) ^a^	Ra (nm) ^b^	Ra/h (%)	T_d_ (°C)	T_g_ (°C)	T (%)
**A0**	215	0.44	0.20	418	266	>90%
**A10**	200	0.56	0.28	427	-	
**A20**	215	0.61	0.28	431	-	
**A30**	220	0.65	0.29	450	-	
**A40**	210	0.75	0.35	461	-	
**B0**	200	0.31	0.15	426	286	>90%
**B10**	210	0.35	0.17	431	-	
**B20**	220	0.41	0.19	440	-	
**B30**	215	0.57	0.27	461	-	
**B40**	215	0.64	0.29	477	-	
**C0**	220	0.43	0.19	405	270	>90%
**C10**	210	0.49	0.23	419	-	
**C20**	220	0.53	0.24	432	-	
**C30**	215	0.60	0.28	443	-	
**C40**	210	0.69	0.33	454	-	

^a^ Thickness of the prepared thin film. ^b^ Ra is the average roughness of the prepared thin films, respectively.

**Table 2 polymers-12-01058-t002:** Summary of dielectric constant and surface energy for various hybrid dielectric films.

No.	Dielectric Constant [-]				Water Contact Angle [^o^]	Diidomethane Contact Angle [^o^]	Surface Energy [mJ·m^−^^2^]
	1 kHz	10 kHz	100 kHz	1 MHz			
**A0**	4.53	4.49	4.43	4.10	79.78	35.39	51.02
**A10**	5.78	5.20	4.86	4.27	80.26	42.65	47.66
**A20**	6.93	6.30	6.11	5.02	80.76	43.09	45.73
**A30**	7.24	7.08	6.75	6.35	83.99	50.53	44.17
**A40**	9.51	8.88	7.47	6.53	78.30	29.23	52.33
**B0**	4.47	4.44	4.41	4.07	78.46	39.10	48.74
**B10**	5.57	5.11	4.72	4.03	80.44	45.49	45.97
**B20**	6.03	6.02	5.98	5.00	81.17	51.38	44.68
**B30**	7.14	7.01	6.72	5.98	84.91	51.83	42.07
**B40**	8.58	7.96	7.20	6.31	76.12	37.04	51.37
**C0**	4.22	4.10	4.01	3.98	77.38	39.24	49.47
**C10**	5.44	5.07	4.69	4.00	78.60	46.55	47.20
**C20**	5.99	5.79	5.69	5.12	78.80	51.27	44.70
**C30**	6.91	6.83	6.59	5.45	82.00	51.38	43.34
**C40**	7.87	7.78	7.01	6.29	74.49	38.22	51.02

**Table 3 polymers-12-01058-t003:** The electrical data of OTFTs fabricated by various hybrid dielectrics.

No	LCD [A·cm^−2^] (at −2 MV·cm^−1^)	V_t_ [V]	μ [cm^2^·V^−1^·s^−1^]	I_ON_/I_OFF_ [-]
**A0**	7.5 × 10^−10^	−2.1	1.81 × 10^−2^	1.13 × 10^3^
**A10**	1.5 × 10^−9^	4.1	7.01 × 10^−2^	3.24 × 10^3^
**A20**	2.7 × 10^−9^	−2.3	1.21 × 10^−1^	5.57 × 10^3^
**A30**	4.8 × 10^−9^	−7.3	2.42 × 10^−1^	9.04 × 10^3^
**A40**	7.7 × 10^−9^	3.2	1.07 × 10^−2^	1.16 × 10^3^
**B0**	6.3 × 10^−10^	3.3	5.04 × 10^−2^	1.51 × 10^4^
**B10**	8.6 × 10^−10^	−1.5	2.09 × 10^−1^	2.24 × 10^4^
**B20**	1.7 × 10^−9^	3.9	4.23 × 10^−1^	3.01 × 10^4^
**B30**	2.5 × 10^−9^	−8.1	8.17 × 10^−1^	4.27 × 10^5^
**B40**	5.8 × 10^−9^	2.2	5.51 × 10^−1^	8.50 × 10^4^
**C0**	5.9 × 10^−10^	4.6	4.81 × 10^−2^	8.13 × 10^3^
**C10**	7.7 × 10^−10^	2.4	1.89 × 10^−1^	1.24 × 10^4^
**C20**	1.6 × 10^−9^	−2.6	3.21 × 10^−1^	2.57 × 10^4^
**C30**	3.4 × 10^−9^	3.8	5.62 × 10^−1^	2.04 × 10^5^
**C40**	8.5 × 10^−9^	2.2	2.07 × 10^−1^	2.16 × 10^4^

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
