# Peer review of "Preparation and Application of Organic-Inorganic Nanocomposite Materials in Stretched Organic Thin Film Transistors"

_polymers, 2020, doi:10.3390/polym12051058_

Round 1
Reviewer 1 Report
The theme of the manuscript (a stretchable gate dielectric for OTFS made of a nanocomposite of a polyimide and TiO2-SiO2 nanoparticles) is of interest. However, I do not recommend it to be published in the present form, requiring major revisions, before a new assessment can be made.
In general terms, a major writing revision is required (there are repetitions, typos, ..).
There are many tables with data that is not discussed in detail in the text. They (or most of them, leaving just the relevant information in the body of the manuscript) should be move to a Supporting Information annex. A similar modification should also be made to the Figures.
In addition the output curves of the transistors, in view of the values of the leakage current, should be present as Supporting information.
Some other (not all) points to be addressed:
Characterization of the TiO2-SiO2 nanoparticles is missing (size, size dispersion and structure).
Some references are missing, for instance with respect to the sentence on line 62 (page 2).
On line 96, page 3, the authors refer that “The as-prepared polyimide and TiO2–SiO2 nanoparticles were separately dissolved in DMAc and butanol solvents… “. This needs to be corrected (do the nanoparticles dissolve in butanol?).
In section 2.3, the authors describe the fabrication of OTFTs on silicon substrates. No reference to the performance of such OTFTs and its relevance for the main aim of the paper is made.
On line 182, page 5, the sentence about the values of Tg is not accurate.
The order of the plots in Figure 2 is not consistent with the caption.
On line 241, page 8, the authors refer to the conjugated polymer layer phase separation. This requires clarification.
The authors sentence on line 257 “ … resulting in more charge traps at the interface and thereby improved performance” needs further clarification.
The authors claim on line 296, “…high LCD values are caused by large values of capacitance ..” requires further clarification.
The values of Vt on Table 3 show an erratic dependence on the nanoparticles load. Do the authors have an explanation for this variation ?
Author Response
Detailed Responses to Reviewers
Journal: Polymers (ISSN 2073-4360)
Manuscript ID: polymers-759806
Title: Preparation and Application of Organic–Inorganic Nanocomposite Materials in Stretched Organic Thin Film Transistors
Authors:Yang-Yen Yu * , Cheng-Huai Yang
Comments and Suggestions for Authors
Reviewer #1
The theme of the manuscript (a stretchable gate dielectric for OTFS made of a nanocomposite of a polyimide and TiO2-SiO2 nanoparticles) is of interest. However, I do not recommend it to be published in the present form, requiring major revisions, before a new assessment can be made.
Comment :
In general terms, a major writing revision is required (there are repetitions, typos, ..).
There are many tables with data that is not discussed in detail in the text. They (or most of them, leaving just the relevant information in the body of the manuscript) should be move to a Supporting Information. A similar modification should also be made to the Figures. In addition the output curves of the transistors, in view of the values of the leakage current, should be present as Supporting information.
Response:
We have added the transfer curves and output curves of the transistors in the supporting information as suggested by reviewer. Please see Page 9-10 in the revised paper and S3 and S4 in the supporting information.
Comment :
Some other (not all) points to be addressed:
Characterization of the TiO2-SiO2 nanoparticles is missing (size, size dispersion and structure).
Response:
We have added the TEM image to show the size and structure of TiO2-SiO2 nanoparticles in the supporting information. Please see Page 3 in the revised paper and S1 in the supporting information.
Comment:
Some references are missing, for instance with respect to the sentence on line 62 (page2). Response:
We have added the important references as suggested by reviewer in the revised paper.
Please see references [33-37] and [42-46] in the revised paper.
Comment:
On line 96, page 3, the authors refer that “The as-prepared polyimide and TiO2–SiO2 nanoparticles were separately dissolved in DMAc and butanol solvents… “. This needs to be corrected (do the nanoparticles dissolve in butanol?).
Response:
Thank you for the comments. We have corrected this sentence to “ the as-prepared polyimide dissolved in DMAc and TiO2–SiO2 nanoparticles dispersed in butanol were mixed and stirred…...”. Please see Page 19 in the supporting information.
Comment:
In section 2.3, the authors describe the fabrication of OTFTs on silicon substrates. No reference to the performance of such OTFTs and its relevance for the main aim of the paper is made.
Response:
In section 2.3, we fabricated a metal-insulator-metal (MIM) capacitors on silicon substrate for the MIM direct current measurement, not for the OTFTs measurement. Therefore, the description, “OTFTs were fabricated on Si substrates…….”, is not correct. We have deleted this description in the revised paper. Please see page 20 in the supporting information.
Comment:
On line 182, page 5, the sentence about the values of Tg is not accurate. The order of the plots in Figure 2 is not consistent with the caption.
Response:
Thanks reviewer to point out it. We have checked and corrected the values of Tg in this sentence and make the plots in fig. 2 is consistent with the captions.
Please see Page 4, 1st paragraph in the revised paper.
Comment:
The authors sentence on line 257 “ … resulting in more charge traps at the interface and thereby improved performance” needs further clarification.
The authors claim on line 296, “…high LCD values are caused by large values of capacitance ..” requires further clarification.
The values of Vt on Table 3 show an erratic dependence on the nanoparticles load. Do the authors have an explanation for this variation ?
Response:
It is known that high-k dielectric layer could cause a low operation voltage and low power consumption. The LCD value is related to the thickness of the hybrid dielectric layer and the pinholes density, because the cross-linking between the inorganic and polyimide can make the dielectric layer structure more dense, and the high-k dielectric can improve the capacitive coupling effect between the gate and active channel layer, which can increase the driving current and reduce the operating voltage.In addition, the surface morphology of the dielectric layer affects the structure of the deposited organic semiconductor layer, which in turn affects the performance of the device. A smooth and pinhole-free surface of dielectric layer is important for the interfacial connection during the deposition of organic semiconductor layers, because a smooth interface reduces the charge scattering sites in the channel. Reducing the current leakage from the dielectric interface has a great influence on the electrical performance of the OTFTs. For high-performance OTFTs, the gate dielectric should have a low LCD and a high dielectric constant, which can provide greater surface charge accumulation and simultaneously reduce the operating voltage.It should be noted that the thickness of the dielectric layer must be carefully controlled to minimize the current leakage without greatly reducing the capacitance. S4 show the output characteristics of the OTFTs using (a) (A0), (b) A30, (c) B30 and (d) C30 as the dielectric layer, respectively. The threshold voltages (Vt) of OTFTs based on hybrid films are very small, so only a small gate voltage is needed to turn on the gate. Surface polarization may result in smaller threshold voltages, which can lead to the filling defects of local carrier. Most of the Vt displacement is affected by three factors, which are the charge defect trapping, surface polarization, and ions. In this study, the variation of vth displacement is very small and within the error range, indicating the addition of different proportions of inorganic nanoparticles on the interface of the dielectric and the semiconductor layer will not cause obvious charge defects, so it will not cause obvious displacement of Vt.
We have added this description in the revised paper. Please see pages 9-10 in the revised paper.
Reviewer 2 Report
Referee Report on Manuscript #polymers-759806 by Yu et al.
General Comments
The paper is an attempt for investigating a hybrid dielectric films comprising polyimide, TiO2-SiO2, and jeffamine D2000 or polyurethane as a gate dielectric material for stretchable organic thin film transistors. The authors provide various investigations and comparisons for finding out optimal composition of elements for the application, however, I recommend the paper will be re-considered after the authors address some major comments I listed as follows.
Major Comments
1. The authors mentioned the inorganic composite of TiO2-SiO2, Jeffamine D2000, and polyurethane have been already investigated for organic thin film transistor in several previous studies. The author should clarify the advantage and advancement of the presented work compared to the relevant previous works.
2. The optical transparency of the whole TFT device should be further verified by also providing its photograph image.
3. Regarding the surface flatness of the presented dielectric films, it should be compared with previous studies in order to clarify its good flatness.
4. Can the increased water contact angle of the film including SiO2-TiO2 result from the surface roughness rather than the number of OH groups? If the effect of the OH groups is dominant, experimental data should be provided.
5. The author mentioned the high dielectric constant of TiO2-SiO2 in a low dielectric constant polyimide matrix results in a surface energy decrease. However, theoretically, there is no relationship between the dielectric constant and surface energy of materials. The authors need to clarify how the high dielectric constant affect the surface energy.
6. The performance of the OTFT in respect to tensile strain should be compared with the previous stretchable transistor devices.
Minor Comments
7. The authors mentioned the presented film was used for an OTFT gate layer although the film was actually used for a gate dielectric layer. The author should revise the expression in order to prevent misleading from readers.
Author Response
Detailed Responses to Reviewers
Journal: Polymers (ISSN 2073-4360)
Manuscript ID: polymers-759806
Title: Preparation and Application of Organic–Inorganic Nanocomposite Materials in Stretched Organic Thin Film Transistors
Authors:Yang-Yen Yu * , Cheng-Huai Yang
Comments and Suggestions for Authors
Reviewer #2
The paper is an attempt for investigating a hybrid dielectric films comprising polyimide, TiO2-SiO2, and jeffamine D2000 or polyurethane as a gate dielectric material for stretchable organic thin film transistors. The authors provide various investigations and comparisons for finding out optimal composition of elements for the application, however, I recommend the paper will be re-considered after the authors address some major comments I listed as follows.
Comment:
The authors mentioned the inorganic composite of TiO2-SiO2, Jeffamine D2000, and polyurethane have been already investigated for organic thin film transistor in several previous studies. The author should clarify the advantage and advancement of the presented work compared to the relevant previous works.
Response:
Compared with the previous works, this study has several innovations and differences, as described below.
It is known that the polyimide (PI) has become one of the most important high-performance polymers in the electronics field due to its excellent thermal/mechanical/electrical properties and dimensional stability. At present, there are no polyimide hybrid materials used in stretched organic thin film transistors, because the commercial PI usually has poor solubility and high melting point due to its rigid polymer backbone and strong intermolecular interactions. Therefore, this limits its further application in thin film-based electronic devices. In addition, PI films usually have a dark color (yellow or brown) due to their highly conjugated aromatic chains and strong charge transfer bonds, which results in poor optical transparency. In this study, PI fluorination can produce better solubility, especially for polyetherimides containing trifluoromethyl substituted structures. This type of PI is usually a dianhydride monomer synthesized by a polycondensation reaction between a fluorine-containing diether diamine monomer and ordinary fluorine. Moreover, because the composed methyl group has a larger free volume, and fluorine atoms have unique physical and chemical properties, such as a larger electronegativity, a smaller atomic radius, and a lower molar polarizability, etc. Therefore, the polyimide used in this study has more ideal performance than traditional PI.
For application of PI in organic thin-film transistors (OTFTs), there are some challenge need to be overcame. OTFTs requires a high operating voltage, which need PI can integrate with other electronic components in practical applications. Currently, there are two main methods to solve this problem. One is to use ultra-thin films as dielectrics; However, ultra-thin films are difficult to manufacture and easily damaged. Another is to use high dielectric constant (k) inorganic materials as the medium, but these high-k materials still have some disadvantages, such as large leakage current and rough surfaces. In this regard, hybrid thin films containing high-k inorganic components and organic materials have become the most promising dielectric materials for OTFTs to reduce operating voltage. PI is regarded as a promising organic material due to its superior properties. So far, a variety of methods have been introduced in the literature to prepare hybrid thin films and the properties of the prepard hybrid thin films can be effectively adjusted by selecting materials and adjusting the content of inorganic oxides. The main research goal is to develop a hybrid film with a smooth surface and low defect density to minimize the leakage current under working conditions and improve the overall device performance.
In this study, we first used 9,9-bis (3-amino-4-hydroxyphenyl) fluoro (BAHPF) and 4,4 '-(hexafluoroisopropylidene) diphthalic anhydride ( 6FDA) to synthesize new PI with high processability. Due to the polyfluoro and hydroxyl functional groups on the main chain, the prepared PI can be dissolved in many common solvents at room temperature, such as N-methyl-2-pyrrolidone, N, N-dimethylacetamide ( DMAc), N, N-dimethylformamide (DMF), chloroform, dichloromethane, tetrahydrofuran (THF), etc. The solubility can reach 20 wt%. Therefore, PI thin film could be easily prepared and exhibited high optical transparency. In addition, the hydroxyl groups can further undergo hydrolysis and condensation reactions with high-k TiO2-SiO2 nanoparticles to prepare the PI/TiO2-SiO2 hybrid materials. In this study, the PI/TiO2-SiO2 hybrid materials have been proved to be the effective dielectric materials in OTFTs. The best performance of OTFTs can be obtained by easily adjusting the amount of TiO2-SiO2 added in the PI matrix and adding Jeffamine D2000 or polyurethane as additives into the dielectric layer, which can simultaneously smooth the surface of the dielectric and passivate its surface defects, thereby further improving the performance of stretched OTFTs.
The above-mentioned description can be found in our previous study. We have added this paper as the reference [43] in the revised paper.
Comment :
The optical transparency of the whole OTFT device should be further verified by also providing its photograph image.
Response:
S2 shows the optical transmittance of A0, A30, B30, and C30 fulms as the gate dielectric of OTFTs device in the visible light region of 400-700 nm. It indicates that the optical transmittances of all samples are greater than 75%.
We have added S2 and the relative description in the Supporting Information. Please see Page 3 and 21 (S2) in the revised paper.
Comment:
Regarding the surface flatness of the presented dielectric films, it should be compared with previous studies in order to clarify its good flatness.
Response:
The previous studies pointed out that when the ratio of surface roughness to thickness is less than 0.5%, the material has a good flatness. The value of roughness/thickness in this study is less than 0.35%, so the prepared dielectric thin film in this study has a better surface flatness than those of studies in literature. We have added this comparison in the revised paper. Please see Page 4 last line and Page 5, lines 1-3 in the revised paper.
Comment:
Can the increased water contact angle of the film including SiO2-TiO2 result from the surface roughness rather than the number of OH groups? If the effect of the OH groups is dominant, experimental data should be provided.
Response:
Thanks reviewer to provide this comment. We agree with the reviewer's opinion that the increase in water contact angle is mainly due to the change in surface roughness rather than the number of OH groups. We have corrected it in the revised paper. Please see Page 9, lines 1-2 in the revised paper.
Comment:
The author mentioned the high dielectric constant of TiO2-SiO2 in a low dielectric constant polyimide matrix results in a surface energy decrease. However, theoretically, there is no relationship between the dielectric constant and surface energy of materials. The authors need to clarify how the high dielectric constant affect the surface energy.
Response:
Thanks reviewer to provide this comment. We agree with the reviewer’s opinion. The present statement will cause misunderstandings. We have corrected it as follows: the addition of high dielectric TiO2-SiO2 in a low dielectric PI matrix can change the surface roughness of the PI/ TiO2-SiO2 hybrid film, which in turn affects the surface energy of the hybrid film. We have corrected it in the revised paper.
Please see Page 9 in the revised paper.
Comment:
The authors mentioned the presented film was used for an OTFT gate layer although the film was actually used for a gate dielectric layer. The author should revise the expression in order to prevent misleading from readers.
Response:
In the revised paper, the “gate layer” and “gate dielectric layer” in the text have been replaced by “gate dielectric” in order to ensure the consistency of the writing of the article as suggested by reviewer.

Round 2
Reviewer 1 Report
Review report on the manuscript: polymers-759806-peer-review-v1_Preparation and Application of Organic–Inorganic Nanocomposite Materials in Stretched Organic Thin Film Transistors – R1
The revised version of the manuscript does show improvements with respect to the last version. However, I do not recommend the publication in the present form.
An additional language revision is needed. Some typos: the mobility units shoud be cm2V-1 s-1 (not a capital S); on line 72 it should read PEDOT and not EDOT; the order of the references appearing in the text should be revised; there are some repeated parts of the manuscript (e.g. text starting at line 147 and at line 229; there is also repetition in the text between lines 152 and 158); the reference numbers are repeated in lines 32, 39, 160; “BT0” in line 336 is in fact B0?; I suggest the current on/off ratio should be indicated as Ion/Ioff;
In line 63, the meaning of PBA and PDCA should be specified;
In line 72 it is mentioned that the gate contact consists on PEDOT:PSS but the scheme in Figure 1 shows a drop of E-GaIn;
In line 75 the authors should detail the meaning of “the fluorine atom had high transmittance”;
In line 146 (and in other parts of the text), instead of “S1” it should read “Figure S1”;
In line 158 the authors refer to the “intermolecular force between polyimide and TiO2-SiO2”, however the scheme in Figure 1 shows a chemical bond;
In line 159 the authors claim that “the chain reaction is restricted and Td and thermal stability …increases”. This needs further correction/clarification. In this part (ending in line 180) there seems to be some confusion between the factors affecting Tb and those affecting Td;
In Figure 2 the order of the various plots is not in agreement with the caption (they should be identified as a, b and c);
A reference should be indicated for the authors conclusions in the new text in lines 191-194;
The discussion about the various devices is confusing. For instance, the authors start by discussion the capacity of the dielectric (up to line 266) and then change the subject to the contact angle, recovering the capacity discussion in line 289. The OTFT results are also repeated in several parts of the manuscript. The authors should rearrange the paper in order to have a continuous line of discussion for the various experiments.
In lines 320 and 322, the authors refer to “oil contact angle”. What is the meaning ?
I disagree with author´s claim in line 332 that the scattering of the Vt results “is very small” and “within the error range”. Table 3 shows variation range of more than 10V.
The values of LCD in Table 3 should specify at which voltage they correspond, to allow a better comparison with the OTFT characteristics. The same applies in line 363;
In line 389, what is the meaning of “OM”?
In line 404, what do the authors mean by “ a new method” ?
Author Response
Detailed Responses to Reviewers
Journal: Polymers (ISSN 2073-4360)
Manuscript ID: polymers-759806-R2
Title: Preparation and Application of Organic–Inorganic Nanocomposite Materials in Stretched Organic Thin Film Transistors
Authors:Yang-Yen Yu * , Cheng-Huai Yang
Comments and Suggestions for Authors
Reviewer #1
The revised version of the manuscript does show improvements with respect to the last version. However, I do not recommend the publication in the present form.
Comment and Response:
Thank you for these comments. We have corrected these typos and rewritten some unclear sentences in the revised paper. The following is our point-by-point response to the reviewers' comments.
Some typos: the mobility units should be cm2V-1 s-1(not a capital S)à cm2V-1s-1 (please see line 309 and 310 in revised paper); EDOTàPEDOT (line 74); the order of the references appearing in the text should be revised: We have corrected the order; there are some repeated parts of the manuscript (e.g. text starting at line 147 and at line 229; there is also repetition in the text between lines 152 and 158); We have deleted repetition in the text (lines 170-173); the reference numbers are repeated in lines 32, 39, 160: We have corrected it; “BT0” in line 336 is in fact B0? Yes, it’s B0. We have corrected it (line 376); I suggest the current on/off ratio should be indicated as Ion/Ioff: We have corrected it (lines, 370, 374, 388 etc.); In line 63, the meaning of PBA or PDCA should be specified; PBA: poly(butyl acrylate) and PDCA: 2,6-pyridine dicarboxamide (lines, 63-64); In line 72 it is mentioned that the gate contact on PEDOT:PSS but the scheme in Figure 1 shows a drop of E-GaIn. We have corrected it. In line 146 (and in other parts of the text), instead of “S1” it should read “Figure S1”: We have corrected it (line 161), and we also corrected S2, S3, and S4 to Figure S2, Figure S3, and Figure S4, respectively; In Figure 2 the order of the various plots is not in agreement with the caption (they should be identified as a, b and c): The (a), (b), (c) have added in Figure 2 in the revised paper; The values of LCD in Table 3 should specify at which voltage they correspond, to allow a better comparison with the OTFT characteristics: The values of LCD were measured at -2V (line 305); The same applies in line 363; In line 389, what is the meaning of “OM”?: OM=optical microscope (line 399); In line 404, what do the authors mean by “a new method”: This is a typos, we have corrected it (lines 413-415) ? In lines 320 and 322, the authors refer to “oil contact angle”. What is the meaning? We have rewritten this sentence to make the meaning of the sentence clearer (lines 334-338).
We also have searched the English editing service to improve our manuscript.
Comment:
In line 75 the authors should detail the meaning of “the fluorine atom had high transmittance”;
Response:
We have added the following statement in the revised paper, “the size of CF group is quite big in PI molecule,, which can cause an increase of free volume and a reduction of interaction between the molecular chains, so as to increase the solubility and transparency for the prepared polyimide–TiO2–SiO2 hybrid films”. Please see lines 77-79 in the revised paper.
Comment:
In line 158, the authors refer to the “intermolecular force between polyimide and TiO2-SiO2”, however the scheme in Figure 1 shows a chemical bond; In line 159 the authors claim that “the chain reaction is restricted and Td and thermal stability …increases”. This needs further correction/clarification. In this part (ending in line 180) there seems to be some confusion between the factors affecting Tb and those affecting Td.
Response:
We have rewritten this sentence, “the thermal decomposition temperature increases with increasing content of TiO2-SiO2 nanoparticles due to the formation of chemical bonding between polyimide and TiO2-SiO2, which can restrict the polyimide chain reaction, and the Td and thermal stability for the hybrid films thus increases as TiO2-SiO2 content increases”. Please see lines 170-173 in the revised paper.
Comment:
The discussion about the various devices is confusing. For instance, the authors start by discussion the capacity of the dielectric (up to line 266) and then change the subject to the contact angle, recovering the capacity discussion in line 289. In addition, the OTFT results are also repeated in several parts of the manuscript. The authors should rearrange the paper in order to have a continuous line of discussion for the various experiments.
Response:
Thanks reviewer provided this comment. We have deleted some repetition about the OTFT results in the text, and in order to make the results and discussion of this article clearer, we have added subsections (3.1~3.4) in the revised paper.
3.1 Analysis of optical and thermal properties
3.2 Analysis of stretching property
3.3 Analysis of surface morphology and surface energy
3.4 Analysis of electrical properties
Therefore, the statement of contact angle and capacity of the dielectric has been removed to subsection 3.3 and 3.4, respectively to eliminate any confusion. Please see subsection 3.3 and 3.4 in the revised paper.
Comment:
I disagree with author´s claim in line 332 that the scattering of the Vt results “is very small” and “within the error range”. Table 3 shows variation range of more than 10V.
Response:
We have corrected the description as suggested by reviewer. The statement has corrected to, “ In this study, the variation of Vt displacement is significant, indicating the addition of different proportions of inorganic nanoparticles on the interface of the dielectric and the semiconductor layer will cause an obvious charge defects and Vt displacement.
Please see lines 343-345 in the revised paper.
Reviewer 2 Report
All the comments have been properly addressed in the revision report.
Author Response
Thanks reviewer.
Round 3
Reviewer 1 Report
This new revised version does show improvements with respect to the previous one, but there still aspects that require authors´attention before acceptance for publication final decision is made.
i) Several typos still remain. For example, in the mobility units, in several places, namely in the abstracts but not only, seconds are indicated as "S" and not "s";
ii) the meaning of the sentence in lines 32-33 is unclear
iii) in line 162, it should read "..the size of the particles of the..."
iv)In line 187, I believe it should be "Td" instead of "Tg"
v)In Figure 3, the word "pristine" in "0%(pristine)" should be removed as this refers to the strain;
vi) the text of lines 261-264 is out of place
vii) The sentence in lines 298-300 needs to be revised
viii) The first sentence in line 303 should read "The electrical data of OTFTs is shown in Table 3"
ix)line 305- LCD was not measured at -2V but at -2MVcm^-1. This seems to correspond to a voltage of -40V
x) in lines 331 and 373 it should read 10^-8 instead of 10^-9 A cm^-2.
xi) In the headings of Table 2, instead of "1k"..it should read "1kHz"....
Author Response
Detailed Responses to Reviewers
Journal: Polymers (ISSN 2073-4360)
Manuscript ID: polymers-759806-R3
Title: Preparation and Application of Organic–Inorganic Nanocomposite Materials in Stretched Organic Thin Film Transistors
Authors:Yang-Yen Yu * , Cheng-Huai Yang
Comments and Suggestions for Authors
Reviewer #1
The revised version of the manuscript does show improvements with respect to the last version. However, I do not recommend the publication in the present form.
Comment and Response:
This new revised version does show improvements with respect to the previous one, but there still aspects that require authors ‘attention before acceptance for publication final decision is made.
- Several typos still remain. For example, in the mobility units, in several places, namely in the abstracts but not only, seconds are indicated as "S" and not "s";
Response:
We have corrected S (second) to s in the mobility unit in the revised paper.
- the meaning of the sentence in lines 32-33 is unclear
Response:
The sentence in lines 32-34 has been rewritten as follows in the revised paper:
Because the performance of an organic thin film transistor (OTFT) [5,6] has not been able to reach the same performance of inorganic transistor, researchers have continued to study the use of various semiconductor materials [7-11] to improve their carrier mobility.
- in line 162, it should read "..the size of the particles of the..."
Response:
The sentence in line 161-162 has been rewritten as follows in the revised paper:
It shows that the size of the particles of the as-prepared TiO2-SiO2 is about 30-40 nm with a spherical morphology.
- In line 187, I believe it should be "Td" instead of "Tg"
Response:
In line 187, Tg is correct. We have rewritten the sentence to avoid any confusion. Please see lines 187-189 in the revised paper.
- In Figure 3, the word "pristine" in "0%(pristine)" should be removed as this refers to the strain.
Response:
We have corrected it as suggested by reviewer. Please see Figure 3 in the revised paper.
- the text of lines 261-264 is out of place
Response:
We have removed the sentence in lines 261-264 to lines 335-337 in the revised paper.
- The sentence in lines 298-300 needs to be revised.
- The first sentence in line 303 should read "The electrical data of OTFTs is shown in Table 3"
Response:
We have rewritten the sentence in lines 298-303, and line 303. Please see lines 297-301 and the table caption of Table 3 in the revised paper.
- line 305- LCD was not measured at -2V but at -2MVcm^-1. This seems to correspond to a voltage of -40V
Response:
We have corrected it. Please see line 303 in the revised paper.
- in lines 331 and 373 it should read 10^-8 instead of 10^-9 A cm^-2.
Response:
We have corrected it. Please see line 328 and 374 in the revised paper.
- In the headings of Table 2, instead of "1k".it should read "1kHz"....
Response:
We have corrected the 1k, 10k, 100k, and 1M to 1kHz, 10kHz, 100kHz, and 1MHz, respectively. Please see Table 2 in the revised paper.
